# A Comprehensive Analysis of Immune Response in Patients with Non-Muscle-Invasive Bladder Cancer

**DOI:** 10.3390/cancers15051364

**Published:** 2023-02-21

**Authors:** Guillermo Celada Luis, Eduardo Albers Acosta, Hortensia de la Fuente, Clara Velasco Balanza, Montserrat Arroyo Correas, Nuria Romero-Laorden, Arantzazu Alfranca, Carlos Olivier Gómez

**Affiliations:** 1Department of Urology, Hospital Universitario de La Princesa, 28006 Madrid, Spain; 2Department of Immunology, Hospital Universitario de la Princesa, 28006 Madrid, Spain; 3Department of Medical Oncology, Hospital Universitario La Princesa, 28006 Madrid, Spain

**Keywords:** bladder cancer, immunotherapy, immune response, T-cells, tumor microenvironment, biomarkers

## Abstract

**Simple Summary:**

A comprehensive characterization of cell subpopulations involved in the immune response against bladder cancer has not been performed so far. In addition, due to the high prevalence, recurrence and progression capacity of non-muscle-invasive bladder cancer (NMIBC), the identification of novel biomarkers of tumor progression and response to therapy is of utmost importance. The detailed analysis of the immune landscape in these patients is highly relevant to anticipating tumor behavior and optimizing diagnosis methods and tumor management. We present here the results of the first detailed characterization of immune cell populations in the normal bladder, tumor samples and peripheral blood from patients with NMIBC. We have found specific immune cell subsets differentially expressed in these samples and identified potential markers of tumor progression and patient outcome in peripheral blood. These findings provide relevant information about the host immune response against bladder cancer and set the basis for novel non-invasive procedures for patient stratification and monitoring.

**Abstract:**

Background. Bladder carcinoma has elevated morbimortality due to its high recurrence and progression in localized disease. A better understanding of the role of the tumor microenvironment in carcinogenesis and response to treatment is needed. Methods. Peripheral blood and samples of urothelial bladder cancer and adjacent healthy urothelial tissue were collected from 41 patients and stratified in low- and high-grade urothelial bladder cancer, excluding muscular infiltration or carcinoma in situ. Mononuclear cells were isolated and labeled for flow cytometry analysis with antibodies aimed at identifying specific subpopulations within T lymphocytes, myeloid cells and NK cells. Results. In peripheral blood and tumor samples, we detected different percentages of CD4+ and CD8+ lymphocytes, monocyte and myeloid-derived suppressor cells, as well as differential expression of activation- and exhaustion-related markers. Conversely, only a significant increase in bladder total monocytes was found when comparing bladder and tumor samples. Interestingly, we identified specific markers differentially expressed in the peripheral blood of patients with different outcomes. Conclusion. The analysis of host immune response in patients with NMIBC may help to identify specific markers that allow optimizing therapy and patient follow-up. Further investigation is needed to establish a strong predictive model.

## 1. Introduction

Bladder cancer has high morbidity and mortality [1], and it represents a significant global public health issue due to the high recurrence rate and progression in localized stages [2,3,4]. Once diagnosed, these patients may require different intravesical treatments, and advanced and metastatic diseases usually need complex surgical and systemic therapies. Therefore, this condition is currently considered a pathology with high consumption of economic resources [5].

Approximately 75% of bladder cancers are confined to the urothelial mucosa [6], which has different prognostic and therapeutic implications than those affecting the bladder muscle. Approximately 90% of bladder tumors are urothelial carcinoma, followed by squamous cell carcinoma (1–7%) and, to a much lesser extent, by adenocarcinoma (0.5–2%) [7].

Transurethral Resection of Bladder Tumor (TURB) is the main approach to diagnosing and treating non-muscle-invasive bladder cancer (NMIBC). The risk of recurrence and progression of NMIBC after the first TURB varies from 48–70% and 7–40%, respectively [8].

It is crucial to determine the risk of recurrence and progression to muscle-invasive disease in the short and medium term. This stratification is determined at the diagnosis, and tumors may be classified into four risk groups: low, intermediate, high, and very high, following the system developed by the European Association of Urology (EAU) (www.nmibc.net; accessed on 15 November 2022). This system includes different variables such as age; if the tumor is de novo or recurrent; its size and numbers; associated carcinoma in situ; the pathological stage according to the TNM classification; and the differentiation degree (low-grade or high-grade) [9,10].

The treatment of low-risk NMIBC varies from different agents, and nowadays, Mitomycin C is widely used in low-grade tumors. On the other hand, Bacillus Calmette-Guérin (BCG) is the treatment of choice for high-grade NMIBC; therefore, both treatments are standard of care.

The interaction of cancer with the immune system promotes the recruitment and activation of different myeloid and lymphoid cell populations, as shown by infiltrates of macrophages, dendritic cells, mast cells, granulocytes, and lymphoid cells found in immunopathological studies [11]. The immune system plays an efficient role in limiting cancer progression at early stages; however, one or more cancer cell variants may survive this initial phase, leading to a dynamic equilibrium wherein tumor-specific lymphocytes exert continuous selective pressure over cancer cell variants. Finally, tumors may develop different immunoevasive mechanisms to escape immune response, which leads to overt tumor development and may modulate cancer response to therapy.

There is an increasing interest in studying the precise mechanisms by which immune cells promote cancer cell death and response to different drugs. Previous data suggest that, in patients with bladder cancer, the content of immune cells in urine is similar to that in the tumor microenvironment, but this issue needs further clarification [12]. On the other hand, although lymphoid infiltrates in some types of melanomas, colorectal, and breast cancers predict a better prognosis [13], this issue has not been studied in detail in bladder cancer. Hence, to our knowledge, no previous studies describe lymphoid populations in these tumors and their correlation with peripheral blood and healthy bladder tissue.

We present here a descriptive study conducted in a population of 41 patients with NMIBC, in which immune cell subpopulations from tumor tissue, adjacent healthy bladder, and peripheral blood have been analyzed. We show the differential distribution of immune cell subsets among the different tissues and provide preliminary evidence for biomarkers predictive of patient outcomes in peripheral blood.

## 2. Materials and Methods

### 2.1. Study Population

This descriptive study was performed in the period between February 2019 and January 2022. Samples from peripheral blood, urothelial bladder cancer and adjacent healthy urothelial tissue (confirmed by a Pathologist) were collected from 41 patients treated at the Department of Urology of La Princesa University Hospital, Madrid, Spain. The epidemiological and clinical characteristics of these individuals are detailed in Table 1. Patients were divided into low- and high-grade urothelial bladder cancer according to WHO/ISUP TNM 8th edition. In order to achieve a more homogeneous sample, we only collected urothelial pTa and pT1 bladder carcinoma samples, excluding CIS, muscle-invasive, and other pathological differentiation-containing samples. Recurrences were identified in cystoscopies performed during patient follow-up. Tumor size was determined by comparing the tumor with the resection loop (2.5 cm long) during surgery. Tissue samples were collected in the operation room (OR) and kept at 4 °C until their processing.

### 2.2. Sample Processing

Blood and bladder samples (both urothelial bladder cancer and normal bladder urothelium were processed in the laboratory as follows:

After disaggregation of fresh samples from tumors and normal bladder urothelium with a razor blade and subsequent filtration through a 40 μm-pore cell strainer (Falcon), mononuclear cells were isolated using a density gradient centrifugation (Ficoll-Paque; Pan-Biotech). Peripheral blood mononuclear cells (PBMC) were isolated as follows: peripheral blood collected in EDTA-treated tubes was diluted 1:1 with PBS and carefully added to Ficoll-Paque (Pan-Biotech; proportion 4:3) in order to avoid mixing. After centrifugation (400× *g*, 30 min, without brake), PBMC-containing interphase was collected with a pipette, transferred to a new tube and washed twice with PBS. Finally, isolated cells were cryopreserved in media containing 90% fetal bovine serum (FBS; HyClone TM) and 10% DMSO (Inilab) until analysis.

### 2.3. Flow Cytometry Analysis

Mononuclear cells from blood and tissues were incubated with Human FcR blocking reagent for 10 min at room temperature and then labeled with the different antibodies for 30 min at 4 °C. Finally, the cells were washed and resuspended in 200 μL PBS 1×. The antibodies were distributed in the following panels:Panel 1 (T lymphocytes 1): CD49b, CD127, CD3, LAG3, CD4, CD25 and CD8;Panel 2 (T lymphocytes 2): TIM3, PD1, CD3, ICOS, CD27, CD4 and CD8;Panel 3 (myeloid cells): MHCII, CD 127a/b (SIRP), IDO1, CD14, CD11c and CD123;Panel 4 (NK cells): KIR, NKG2a, NKG2d and CD56.

Fluorochromes and providing Companies are listed in Appendix A. Samples were acquired using a FACSCanto II flow cytometer (BD Biosciences), and data obtained were subsequently analyzed with FlowJo software (BD Biosciences).

### 2.4. Statistical Analysis

Quantitative variables obtained by flow cytometry were represented in bar graphs as mean ± standard deviation (SD) or in box-and-whiskers plots. Analysis of variance was performed to determine differences in cell subsets between blood and tissues (tumor and normal bladder urothelium), and Sidak’s multiple comparison post-test was used. For correlation analyses, Spearman’s coefficient r was calculated. Differences between the 2 groups (Ta vs. T1, relapse vs. non-relapse, and low grade vs. high grade) were determined using an unpaired *t*-test. We used a multivariable logistic regression model to assess the association of the expression of molecules in the different leukocyte populations with TNM independently of known risk factors (e.g., differentiation and tumor size). Variables included in the model were those found to be statistically significant, with *p* < 0.1 in the univariable analysis. Data analyses were performed using Graph Pad Prism 8 Software (GraphPad Software, San Diego, CA, USA, www.graphpad.com, accessed on 15 November 2022) and Graph Stata (StataCorp, College Station, TX, USA).

### 2.5. Ethical Approval

The study has been approved by the institutional review board and ethics committee at La Princesa University Hospital (IRB00005840; register number 4709). It meets international standards of data protection and is in line with practice guidelines established in the Declaration of Helsinki. Informed consent was signed by all the patients before surgery.

## 3. Results

### 3.1. Differential Immune Profile in Samples from Patients with Urothelial Bladder Cancer

In order to carry out a detailed characterization of immune cell populations involved in anti-tumor response to urothelial bladder cancer, we analyzed by flow cytometry the percentage of different immune cell subsets in peripheral blood, healthy bladder tissue and tumor samples from a cohort of 41 patients (Table 1).

When comparing peripheral blood and tumor infiltrate (Figure 1A), we found a significant increase of CD4+ lymphocytes and monocytes in peripheral blood, while the percentage of CD8+ T lymphocytes and myeloid-derived suppressor cells (MDSCs) was significantly higher in tumors. Further analysis of specific activation- and exhaustion-related markers within these populations showed a significant increase of CD4+ CD127+ lymphocytes, CD4+ CD27+ lymphocytes, CD123+ monocytes, IDO+ monocytes and SIRP+ monocytes in peripheral blood. Conversely, higher percentages of CD8+ CD127+ lymphocytes, CD8+ LAG3+ lymphocytes, CD8+ PD-1+ lymphocytes and IDO+ MDSCs were found in tumor samples.

Comparative analysis of samples from peripheral blood and healthy bladder tissue (Figure 1B) showed higher amounts of total CD4+ lymphocytes, CD4+ CD127+ lymphocytes and CD4+ CD27+ lymphocytes in the blood, while increased percentages of total CD8+, CD8+ CD49b+, CD8+ CD127+, CD8+ LAG3+, CD8+ CD27+, CD8+ PD-1+ and CD8+ TIM3+ lymphocytes were detected in the bladder. Finally, only a significant increase in bladder total monocytes was found when comparing bladder and tumor samples (Figure 1C).

We next determined the expression levels of different surface markers present in immune subpopulations by quantifying their mean fluorescence intensity (MFI) in flow cytometry analyses. When comparing peripheral blood and tumor samples, significantly higher expression of CD127 and LAG3 within CD8+ lymphocytes, and increased IDO within MDSCs, were found in the tumors. On the other hand, higher levels of SIRP in monocytes were identified in peripheral blood (Figure 2A). Furthermore, a strong significant correlation was observed in CD127 MFI (r = 0.82; *p* < 0.0001) and in LAG3 MFI (r = 0.76; *p* < 0.0001) of CD8+ lymphocytes between tumor and peripheral blood. Likewise, a significant correlation was found in monocyte SIRP MFI (r = 0.49; *p* = 0.017) between both samples.

Furthermore, when considering healthy bladder and peripheral blood, we detected significantly increased expression of LAG3 in CD4+, CD127, and LAG3 in CD8+ bladder lymphocytes. Similarly, the expression of IDO in both monocytes and MDSCs was higher in bladder tissue. Only SIRP expression in monocytes was significantly lower in bladder samples than in blood (Figure 2B).

Interestingly, when comparing bladder tissue and tumor samples, increased levels of LAG3 were seen in CD4+ lymphocytes, CD8 and CD127 in CD8+ lymphocytes, as well as SIRP within the monocyte population was found in the bladder (Figure 2C).

### 3.2. Immune Markers in Tumor Progression and Patient Outcome

In view of the differences observed in the immune profile among samples, we sought to investigate the differential expression of immune markers in relation to parameters of tumor evolution and patient outcome (i.e., relapse appearance, TNM—pTa/pT1 -, differentiation –low/high grade- and tumor size—</> 3 cm -). As shown in Figure 3A, a higher percentage of CD4+ TIM3+ cells were detected in the peripheral blood of patients that do not present relapse. In addition, monocytes in peripheral blood from patients with pTa tumors showed lower IDO levels than those of patients that bear pT1 tumors (Figure 3A). Likewise, the percentage of different immune cell subsets in bladder and tumor samples showed significant differences in patients with and without relapse, tumors with low and high histological grade, and pTa/pT1 tumors. These cell populations are detailed in Figure 3B.

### 3.3. CD4+ CD27+ Cells and IDO+ Monocytes Are Independent Markers for Tumor Progression

In order to determine whether previously analyzed immune markers could be used as independent markers for tumor progression or patient outcome, a multivariable logistic regression analysis was performed considering relevant variables such as histological grade and tumor size. This analysis showed that the percentage of CD4+ CD27+ cells and IDO+ monocytes remained independent predictors of tumor progression (Figure 4A). This model, which combines immune parameters of patient peripheral blood with histological grade and tumor size, discriminates pTa and pT1 tumors with a sensitivity of 75%, a specificity of 96.65%, a positive predictive value of 90%, and a negative predictive value of 88% (area under ROC curve is 0.9783) (Figure 4B).

## 4. Discussion

Many studies characterize different aspects of anti-tumor immune response in various types of cancer. Thus, in non-small cell lung cancer, a higher density of CD8+ tumor infiltrate lymphocytes (TILs) may promote immune response at the level of the tumor microenvironment [14]. Likewise, a study in patients with metastatic colon cancer determined the presence of CD8+ population densities and T lymphocyte infiltrates through immunohistochemistry in samples from metastatic lesions (lung and liver), demonstrating the prognostic significance of the existence of CD8+ in these tumors [15]. On the other hand, myeloid-derived suppressor cells (MDSC) have been detected in the blood of patients with glioblastomas [16] and other tumors, with a significant increase in the early and advanced stages of the disease [17]. Increasing data support the immunosuppressive role of this group of cells through their interaction with adaptive and innate immunity, as well as through stimulation of angiogenesis.

Chronic stimulation of T cells by tumor antigens leads to a process of lymphocyte exhaustion, accompanied by increased surface LAG3 and PD1 levels [18]. LAG3 reduces T cell proliferation, decreases the secretion of specific cytokines, and contributes to regulatory T cell-mediated immune suppression [19]. The presence of LAG3 in TILs from ovarian cancer has been previously reported [20]. Little is known about the role of CD8+ CD127+ lymphocytes, although their presence in follicular lymphoma may have a favorable prognostic role by effectively promoting the immune response in the tumor [21].

Most of the current evidence on the immune response to bladder cancer focuses on murine tumor models, on determining global lymphoid populations through immunohistochemistry in tissue, or on studying immune profiles in urine in patient samples. Some studies performed by immunohistochemistry techniques have demonstrated that high levels of dendritic cell infiltrates were associated with a worse response to BCG [22]. Similarly, both low CD8+/MDSC and CD8+/TReg ratios are related to poorer outcomes in bladder cancer [23]. Likewise, patients with high levels of CD8+ exhaustion markers have a higher risk of recurrence after BCG [24]. A recently published study analyzes different lymphocyte populations in urine before and after BCG treatment by mass cytometry, establishing possible markers of response and selection of therapy in patients with NMIBC [25].

Our study presents for the first time a detailed analysis by flow cytometry of the immune infiltration profile in tumor tissue and its correlation with healthy bladder tissue and peripheral blood in patients with NMIBC. Due to the high prevalence and recurrence rate of NMIBC, we have focused primarily on this entity independently of muscle-invasive cancers. Likewise, we have not included CIS-containing NMIBC samples because the coexistence of CIS radically modifies the prognosis of the disease. However, in view of the results obtained and the statistically significant differences achieved in NMIBC, it will be of great interest to extend the study to T2 and CIS.

We have found differences in specific immune subsets and differentially expressed markers between peripheral blood and tumor, while similar changes can be observed between peripheral blood and normal bladder urothelium. Interestingly, CD127 and LAG3 expression in CD8+ lymphocytes, and SIRP expression in monocytes, show a correlation between tumor samples and blood, indicating that peripheral expression of these parameters reliably reflects the composition of tumor infiltrate. Of note, few significant differences are found when comparing tumor and healthy bladder (Figure 1). Particularly striking is the fact that, in patients with high-grade tumors, an increase in populations with immunosuppressive characteristics (IDO+ and SIRP+ monocytes) is observed in the healthy bladder in contrast to tumor tissue (Figure 3). These findings highlight the relevance of bladder immune response in the pathogenesis of NMIBC and support the hypothesis that bladder cancer is a pan-urothelial disease.

In our univariable analysis, we observed a significant increase in the percentage of CD4+ TIM3+ in the peripheral blood of patients who have not relapsed. Similarly, we have observed increased CD8+ lymphocytes expressing TIM3 and PD1 exhaustion-related markers in normal bladder urothelium from non-relapsed patients (Figure 3). The expression of PD-1 and TIM3 has been described in exhausted CD4+ and CD8+ lymphocytes; however, activated T lymphocytes up-regulate PD-1 expression, and some subpopulations of activated CD4+ also express TIM3 [26,27]. Therefore, further studies with a larger number of patients are needed to determine the functional role of these immune cells in NMIBC and confirm their predictive role of low probability of relapse. The identification of a marker of tumor evolution in peripheral blood will be of special relevance to improve patient monitoring through non-invasive procedures.

We have obtained a predictive model for TNM which combines peripheral blood parameters (% CD27+ CD4+ lymphocytes, % IDO+ monocytes, LAG3 MFI in CD4+ lymphocytes) with histological grade and tumor size and discriminates pTa and pT1 tumors with a sensitivity of 75% and specificity of 96.65%. This model may help to reliably stratify patients with NMIBC in uncertain cases as an extra tool for patient follow-up and management.

As a limitation of our study, we have not assessed the presence of lymphovascular invasion that may affect tumor-peripheral blood recirculation of immune cells since Pathologists in our hospital only describe the presence or absence of lymphovascular invasion in radical cystectomy but not in transurethral resection samples. Another limitation is that we have conducted a descriptive study, not focused on monitoring the immune response throughout the natural progression of cancer. Therefore, we have not detected any patients with progression. This is an essential aspect that will be addressed in future research.

## 5. Conclusions

Determining the immune marker profile in peripheral blood and tissue may provide relevant information on tumor pathogenesis and have diagnostic and prognostic implications in NMIBC. In this regard, we have detected an increase in populations with immunosuppressive characteristics in the healthy bladder of patients with high-grade tumors. These findings highlight the relevance of bladder immune response in the pathogenesis of NMIBC and support the hypothesis that bladder cancer is a pan-urothelial disease. In addition, in a highly prevalent disease like NMIBC, this type of study is of paramount importance since it can provide early information about the immune response in the tumor microenvironment. The use of a panel of antibodies to assess immune populations in peripheral blood by flow cytometry is feasible, minimally invasive and reproducible in clinical practice. Therefore, it may facilitate better follow-up and decision-making in the diagnosis and treatment of these patients. Despite the heterogeneity in the study population, we still observe significant differences among diverse immune parameters and provide preliminary predictive models that may help to stratify patients. Thus, our findings set the basis for further analyses of immune cell subsets in specific scenarios of NMIBC, which allow us to reliably identify biomarkers and establish predictive models of disease progression.

## Figures and Tables

**Figure 1 cancers-15-01364-f001:**
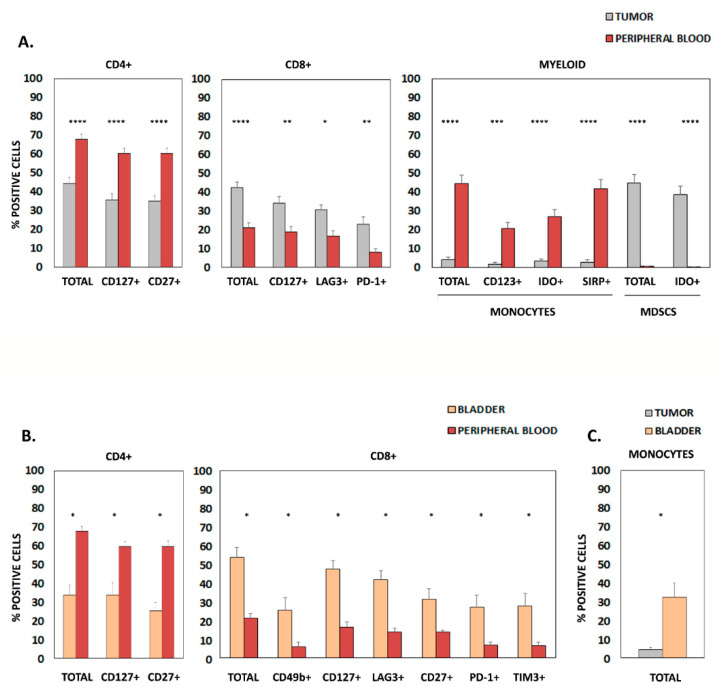
Markers differentially expressed in samples from patients with urothelial bladder cancer. (**A**) Graphics show differentially expressed surface molecules in CD4+ lymphocytes, CD8+ lymphocytes, and monocytes from patient tumor samples and peripheral blood. (**B**) Differentially expressed surface molecules in CD4+ lymphocytes, CD8+ lymphocytes, and monocytes from patient bladder tissue and peripheral blood. (**C**) Differentially expressed markers in monocytes from patient bladder tissue and tumor samples. In all cases, only statistically significant results (Mean + SD of % positive cells) are shown; * *p* < 0.05; ** *p* < 0.01; *** *p* < 0.001; **** *p* < 0.0001. MDSC, myeloid-derived suppressor cells.

**Figure 2 cancers-15-01364-f002:**
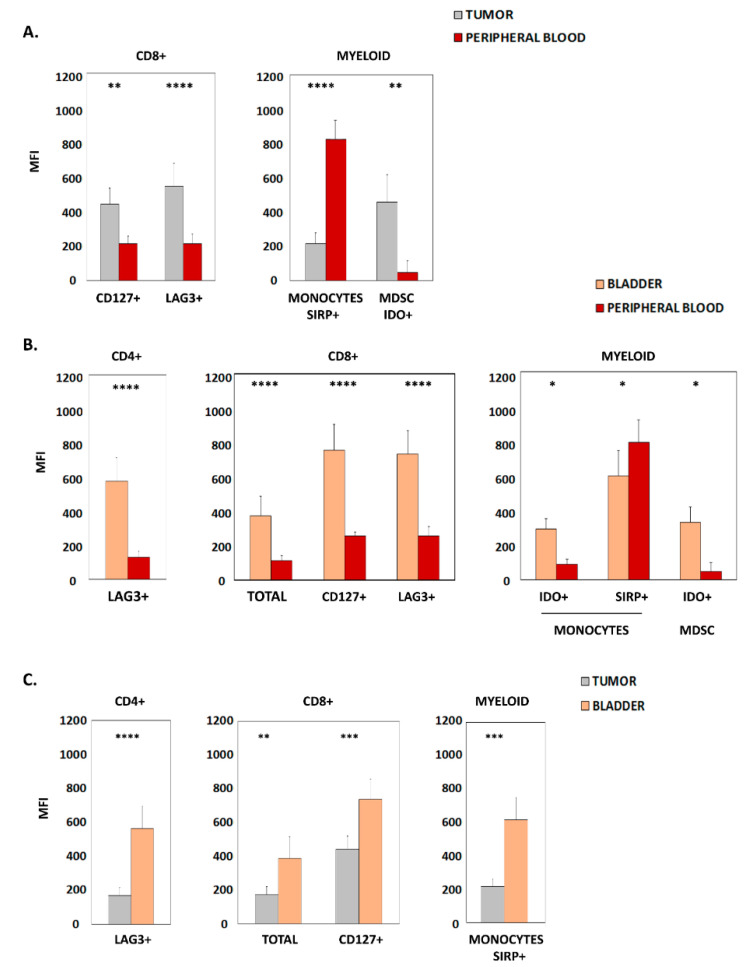
Markers with differential levels of expression in samples from patients with urothelial bladder carcinoma. (**A**) Graphics show the mean fluorescence intensity (MFI) of surface molecules expressed in CD8+ lymphocytes and myeloid cells from patient tumor samples and peripheral blood. (**B**) Graphics show the MFI of surface molecules expressed in CD4+ lymphocytes, CD8+ lymphocytes and myeloid cells from patient bladder tissue and peripheral blood. (**C**) Graphics show the MFI of surface molecules expressed in CD4+ lymphocytes, CD8+ lymphocytes and myeloid cells from patient bladder tissue and tumor samples. In all cases, only statistically significant results (Mean + SD of MFI) are shown; * *p* < 0.05; ** *p* < 0.01; *** *p* < 0.001; **** *p* < 0.0001. MDSC, myeloid-derived suppressor cells.

**Figure 3 cancers-15-01364-f003:**
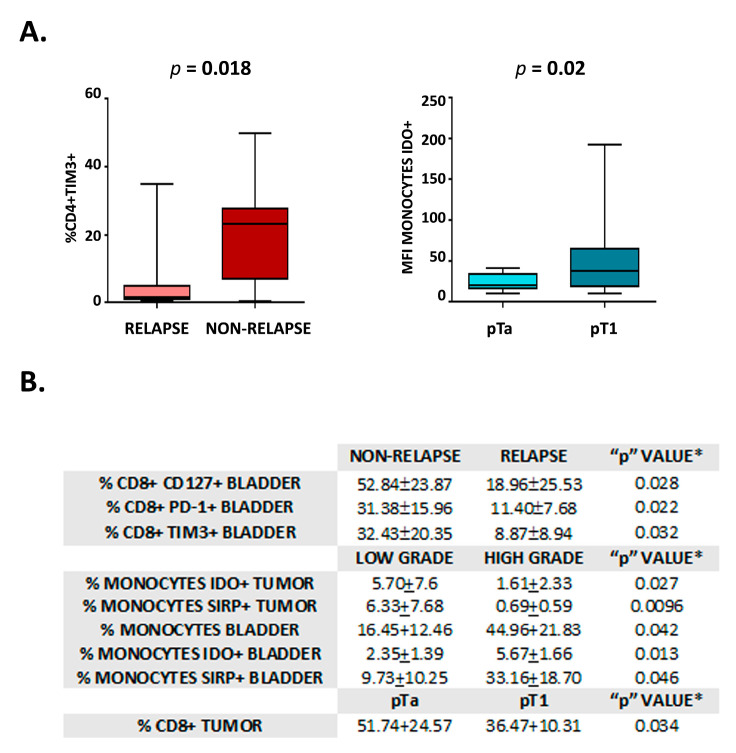
Analysis of immune markers for tumor evolution and patient outcome. (**A**) Graphics show the percentage of CD4+ TIM3+ lymphocytes in the peripheral blood of individuals with Non-relapse vs. Relapse (**left**) and the percentage of monocytes IDO+ in the peripheral blood of individuals with pTa vs. pT1 (**right**). Statistical significance is shown (*p*). (**B**) Table showing the statistically significant results of the univariable analysis to compare individuals with either Non-relapse/Relapse, Low grade/High-grade tumors, and pTa/pT1 tumors. Values correspond to mean ± SD, * student *t*-test.

**Figure 4 cancers-15-01364-f004:**
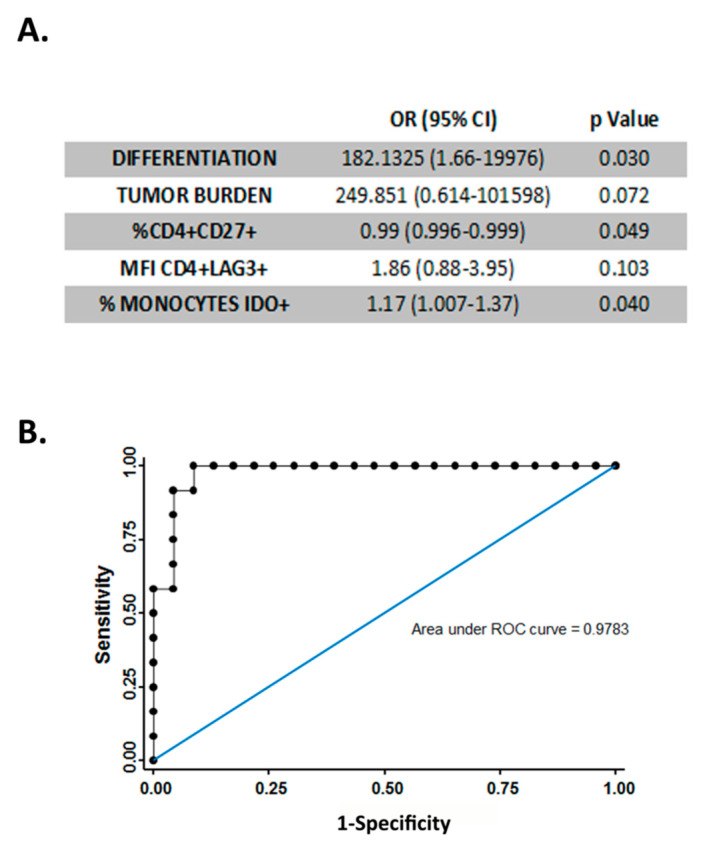
Multivariable analysis. (**A**) Table showing multivariable logistic regression to compare individuals with pTa and individuals with pT1 tumors. All covariates with a *p*-value < 0.1 in the univariable analysis were considered. (**B**) ROC curve showing the ability of the parameters shown in (**A**). to discriminate between patients bearing pTa and pT1 tumors. Sensitivity 75.00%, specificity 95.65%, positive predictive value, 90.00%, and negative predictive value, 88.00%.

**Table 1 cancers-15-01364-t001:** Demographic and clinical characteristics of the study population.

	Patients
n	41
Sex	Female 11 (25.9%) Male 39 (73.1%)
Age	Mean ± SD: 74.97 ± 38.89 (39–94)
WHO/ISUP TNM 8th edition	pTa 27 (65.8%) pT1 14 (34.2%) High grade 25 (60.9%) Low grade 16 (39.1%)
Recurrences at 4 years	Yes 5 (12.2%) (22, 6, 11, 5 and 4 months after diagnosis) No 36 (87.8%)
Tumor Size	≥3 cm: 13 (31.7%) <3 cm: 28 (68.3%)
Risk factor: tobacco	Smokers: 7 (17%) Nonsmokers: 12 (29.4%) Ex-smokers: 22 (53.6%)

## Data Availability

The data presented in this study are available on request from the corresponding author.

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
