# Peer review of "A Comprehensive Analysis of Immune Response in Patients with Non-Muscle-Invasive Bladder Cancer"

_cancers, 2023, doi:10.3390/cancers15051364_

Round 1

Reviewer 1 Report (Previous Reviewer 1)

1. bacille Calmette-Guérin → Bacillus Calmette–Guérin

2. IRB number should be added and authors should make ‘Ethical approval section’ in M&M.

3. 'M&M answer 2'  should be added in M&M section.

4. "M&M answer 4, 5'  should be added in Discussion section (as limitation of study).

Author Response

Reviewer 2 Report (Previous Reviewer 2)

The authors have revised the manuscript and resubmitted to the journal. The response letter was not availble for me. However, the authors have revised the manuscript according to my suggestions.

Author Response

This manuscript is a resubmission of an earlier submission. The following is a list of the peer review reports and author responses from that submission.

Round 1

Reviewer 1 Report

♦ Introduction section

1. Second paragraph (Tobacco~) should be erased.

2. Abbreviation for BCG is incorrect.

3. The introduction section is too long and unpolished. Emphasize only the important points.

♦ M&M section

1. Please provide IRB number.

2. Among 41 patients, were there any patients who showed glandular differentiation

or squamous differentiation, or Etc?

3. Did the authors only visually confirm the normal bladder or pathologically confirm that it is completely cancer-free?

4. Where there any differences in immune markers according to lymphovascular invasion?

5. (Major concern) You said that 5 patients had a recurrence. Were there any patients who showed progression? If so, subgroup analysis should be provided.

Furthermore, Were there any comparison of TNM stage, differentiation, tumor size, etc. between the tumor recurrence and non-recurrence groups at the time of initial diagnosis?

6. Please elaborate more details in PBMC isolation process in section 2.2.

7. Please indicate the media used in PBMC preservation with FBS and DMSO.

8. In Table 1, there are only 5 patients with recurrence. Looking at the graph in Figure 3A and the SD value in Figure 3B, it seems that there is an abnormally different case from the other average values in the relapse group. Could this be a bias?

♦ Results

OK

♦ Discussion

1. A major weakness of this study is that authors did not to analyze Ta + T1 vs T2 or CIS. In fact, Ta and T1 might be similar. Analysis of patients with T2 or CIS may be more useful.

2. The concluding section is too weak. You need a detailed explanation of how the findings from this study may be applied in the future.

Reviewer 2 Report

In this study, the authors have analyzed the immune response of patients diagnosed with non-muscle-invasive bladder cancer (NMIBC). They have demonstrated immune cell populations in normal bladder, tumor samples and peripheral blood from patients with NMIBC. These results provide relevant information about host immune response against BC, and lay a solid foundation for novel non-invasive procedures for patient stratification and monitoring. In this study, 41 patients were involved and analyzed and multiple immune markers were analyzed. The study is of potential interest. I have some comments here.

1. The authors have applied commonly used immune markers for detection in this study. What is the highlight of this study? Or what is different from previous studies?

2. As single-cell sequencing technique has become more and more popular in cancer research, is there any public datasets for NMIBC? The scRNA-seq might provide more information for immune microenvironment.

3. In Figure 2, the authors showed markers with differential level of expression in samples from patients with urothelial bladder carcinoma. They showed data from tumor tissue and peripheral blood. What is the correlation between these markers in tumor tissue and peripheral blood? Was spearman or pearson correlation analysis applied?